# High Mechanical Properties of Stretching Oriented Poly(butylene succinate) with Two-Step Chain Extension

**DOI:** 10.3390/polym14091876

**Published:** 2022-05-04

**Authors:** Xun Li, Min Xia, Xin Dong, Ren Long, Yuanhao Liu, Yiwan Huang, Shijun Long, Chuanqun Hu, Xuefeng Li

**Affiliations:** 1School of Materials and Chemical Engineering, Hubei University of Technology, Wuhan 430068, China; lixun.hubu@foxmail.com (X.L.); dx2418483912@163.com (X.D.); lr302842520@163.com (R.L.); a1361856356@163.com (Y.L.); yiwanhuang@hbut.edu.cn (Y.H.); longshijun.hp@163.com (S.L.); 2School of Materials Science & Engineering, Beijing Institute of Technology, Beijing 100081, China; xminbit@bit.edu.cn

**Keywords:** poly(butylene succinate), chain extension, fracture toughness, flaw sensitivity, crystal morphology

## Abstract

The structure, morphology, fracture toughness and flaw sensitivity length scale of chain-extended poly(butylene succinate) with various pre-stretch ratios were studied. PBS modification adopted from a multifunctional, commercially available chain-extension containing nine epoxy groups (ADR9) as the first step chain extension and hydroxyl addition modified dioxazoline (BOZ) as the second step. Time-temperature superposition (TTS) studies show that the viscosity increased sharply and the degree of molecular branching increased. Fourier transform infrared spectroscopy (FT-IR) confirm successful chain extension reactions. The orientation of the polymer in the pre-stretch state is such that spherulites deformation along the stretching direction was observed by polarized light optical microscopy (PLOM). The fracture toughness of sample (*λ*_fix_ = 5) is *Γ* ≈ 10^6^ J m^-2^ and its critical flaw sensitivity length scale is *Γ*/*W*_c_ ≈ 0.01 m, approximately 5 times higher than PBS without chain-extension (*Γ* ≈ 2 × 10^5^ J m^-2^ and *Γ*/*W*_c_ ≈ 0.002 m, respectively). The notch sensitivity of chain-extended PBS is significantly reduced, which is due to the orientation of spherulites more effectively preventing crack propagation. The principle can be generalized to other high toughness material systems.

## 1. Introduction

Poly(butylene succinate) (PBS) is a commercially available, synthetic polyester possessing many advantages such as biodegradability, renewability, and biocompatibility. The extensive application of the polymers can not only mitigate the negative effect of nondegradable plastics on the environment, but also reduce the dependence on fossil resources [1,2,3]. PBS is a semicrystalline aliphatic polyester that has been gaining attention as a material for developing sustainable solutions oriented towards the worldwide environmental problem of white pollution caused by traditional nondegradable plastics [4,5]. The linear molecular chain structure of PBS contains two carbonyl groups and one butyl group, which leads to the high crystallinity of PBS. PBS’s high-crystallinity paired with its malleability make it widely used in film products [6,7]. PBS has a relatively low molecular weight and its low ductility cannot meet the requirements of some extrusion product applications such as film, monofilament, sheet, strip, sectional material, etc. [8] A general strategy toward the design of anti-fracture polyester material has remained a critical need and a central challenge for long-term applications of polyester in renewable and biodegradable applications [9]. Thus, the method of chain extension has been used to improve its physical properties, especially its toughness and anti-fracture performance. Simultaneously increasing the melt viscosity and toughness to broaden the application range of PBS materials is important. At present there are methods of chemical modification used to solve this problem in PBS. Chemical modifications include forming a branched chain structure on the molecular chain by reacting the chain extender with the end group on PBS (PBS–OH or PBS–COOH), so as to improve the molecular weight and mechanical toughness of PBS. Therefore, a large number of researchers are devoting their efforts to exploring effective PBS chain extenders. In recent years, chain extension agents used for polyester chain extension mainly contain epoxides [10], acid anhydrides [11] or isothiocyanates [12]. In some chain extension reaction processes the chain end carboxyl polymer is extended, while others extend the chain end hydroxyl polymer. Using 1,6-hexmethylene diisocyanate (HDI) as a chain extender, Zhang et al. [13] successfully synthesized multiblock co-polyester, which greatly increased the molecular weight of PBS and excellent mechanical properties were obtained. Zhang et al. [14] focused on the use of toluene-2,4-di-iso-cyanate (TDI) as a chain extender to modify PBS and improved its molecular weight. Zou et al. [15] used 2,2′-bis (2-oxazoline) as a chain extension functional end group. The researchers found that the molecular weight and viscosity of PBS were increased and the crystallization property of PBS was controlled via chain extension. However, although a single chain extender can increase the molecular weight of PBS, it may cause the processing disadvantage of enhanced gel fraction via cross-linking. Effects of orientation and material isotropy on the exact elastic field are necessary design considerations because strength, toughness, and all other mechanical properties of any orthotropic depend on its orientation structure [16]. Under loading progress, the stress and displacement components of any orthotropic composite vary with the variation of material orientation. Stress-orientation analysis of composite structures is carried out using different analytical and semi-analytical approaches where loading is applied along the parallel plane of the composite [17]. Due to the simple structure of PBS, the flexibility of the molecular chain and the relaxation time of the chain segment lead to easy orientation of the chain. Monakhova et al. [18] investigated the mechanical properties of PBS using plane orientation. The researchers found that the addition of a low content of Al_2_O_3_ and TiO_2_ micro- and nanoparticles resulted in a twofold increase in the elastic modulus of the composite, and found the phenomenon of crack propagation. Although studies on polymer chain orientation are rather abundant, the stretching orientation of chain-extended PBS and the significant effect of orientation structure on flaw sensitivity are rarely reported. The flaw sensitivity of an elastic PBS can be estimated by the critical length scale *Γ*/*W*_c_ [19], where *Γ* is the fracture toughness and *W*_c_ is the work to rupture measured with no or negligible flaw. The flaw sensitivity and fracture toughness *Γ* of PBS can be effectively improved by simple pre-stretching in chain-extended compounds. Here, we achieve crack propagation in PBS through the vertical tensile of polymer chains. The aligned polymer spherulites induce anisotropy, making the PBS mechanically weaker between the chains due to the low crystallinity, but stronger along the chains due to the preferred orientation of its spherulites. When the polymer is loaded along the aligned direction, pre-existing flaw deflects from its initial direction of propagation run along propagation of the tensile direction, peel off the material and protects the remaining polymer sample [19]. To demonstrate this principle, we prepared a series of modified PBS samples via pre-stretching progress with two-step chain extension. The goal of the research was to improve the toughness performance of PBS via a two-step chain extension using modified polyfunctional epoxy (ADR9) and2,2′-(1,3-phenylene)-dioxazoline (BOZ). We recently developed the two-steps chain extension method utilized ADR9 as the first-step chain extender and BOZ as the second-step entrapping polybutylene terephthalate (PBT) formation of the branched structure to enable excellent mechanical properties and damp-heat aging resistance [20]. In this work, we further improved the mechanical properties of PBS under optimal conditions. On the basis of two-steps chain extension via two different PBS chain extenders, carboxyl addition (ADR9) and hydroxyl addition (BOZ), the transformation of spherulite morphology was studied by uniaxial stretching to different pre-stretch ratios (degree of orientation) at 95 °C (the oriented structure is stable after cooling). Finally, orientated samples with different pre-stretch ratios were obtained and the morphology, crystallinity, and mechanical properties of the PBS samples were investigated by different techniques. Analysis of the integrated performance of PBS is performed and evaluated to determine potential future applications. 

## 2. Materials and Methods

### 2.1. Materials

PBS (1020MD) (MFI = 25 g 10 min^−1^ at 140 °C, 2.16 kg) was supplied by Showa Denko, Japan. Commercially available ADR-4468 chain extender containing nine epoxy groups (ADR9) with a molecular weight of 6800 g mol^−1^ was supplied by BASF Corporation, USA. 2,2′-(1,3-phenylene)-dioxazoline (BOZ) as a hydroxyl chain extender was supplied by Tiexi Aicheng Industrial Development Co., Ltd, Shanghai, China. Phenol (CP grade) and 1,1-2,2-tetrachloroethane (CP grade) were provided by Shanghai Sinopharm Groups.

### 2.2. Preparation of Two-Step Chain-Extended and Pre-Stretched Sample

All PBS granules were pre-dried in a vacuum oven at 65 °C for 8 h in order to reduce the possibility of hydrolytic degradation. The PBS and chain extenders were well mixed before adding to a rheometer manual-mixer (RM-200C, Harbin, China). The chain-extended PBS was prepared by melt mixing with ADR9 and BOZ with a rotation speed of 30 rpm. The first step of the PBS (50 g) chain extension modification by ADR9 (0.4 or 0.6 wt%) fed into a rheometer manual-mixer at 140 °C and allowed to react for 3 min and thus prepared are denoted as PBS_A0.4_ or PBS_A0.6_. And then the second chain extender BOZ (0.6 or 0.8 wt%) was fed into the mixer containing the PBS_A0.6_ sample. The two-step chain extension sample was obtained and denoted as PBS_A0.6B0.6_ or PBS_A0.6B0.8_ after reacting for 10 min. Afterward, the prepared chain extension samples were injected into plastic molds 75 mm (length) × 4 mm (width) × 1 mm (thickness) in order to obtain the dumbbell-shaped splines, and then uniaxial stretched at 95 °C and pre-stretch ratio (*λ*_pre_ = 1, 3, 5, 7 and 9) prior to unloading. It is not necessary to anneal in order to ensure the crystal structure integrity of the samples. The pre-stretch ratio (*λ*_pre_) is defined by *L*_1_/*L*_0_, here *L*_0_ =20 mm is the initial length in the sample. After pre-stretching, the length of the sample changed to *L*_1_. The samples were released from the extension device to the free state, accompanied by a length change from *L*_1_ to *L*_2_. *L*_2_ was measured after release for 2 days. Compared to the initial state, a fixed stretch ratio exists at the free state, defined as *λ*_fix_ = *L*_2_/*L*_0_ and shown in Table 1. Significantly, all pre-stretching (*λ*_fix_ = 1, 2, 3…9) samples are based on the two-step chain-extended PBS (the PBS _A0.6B0.8_ sample).

#### 2.2.1. Mechanical Properties

*Mechanical testing*. Mechanical tests were carried out on dumbbell-shaped samples with the standard tensile machine (GMT4000, Yangzhou, China) with a 10000 N load cell at room temperature. The initial length *L*_0_ between the two clamps of the tester was 20 mm and the tensile deformation was performed at a series of stretch velocities *ν* from 20 to 200 mm min^−1^, yielding a stretch rate *έ* = *ν*/*L*_0_ from 0.017 to 0.17 s^−1^ [21]. The nominal stress *σ* was estimated from the stretch force divided by the cross-section area of the undeformed sample. Unless otherwise specified, all the testing samples were prepared as dumbbells 75 mm in length, 4 mm wide and 1 mm thick, measured individually for different sample types (e.g., under different *λ*_fix_). In the test, *W*_f_ was defined as the essential work of fracture. The essential work of fracture represents the work at the break region in the integral area of stress–stretch curve. 

*Fracture toughness measurement*. Crack propagation of the PBS samples was measured following the notched stretch. The PBS samples were pre-cut by a razor blade with an initial crack 0.2 times its width and then stretched under a strain rate of 0.008 s^−1^. Fracture toughness *Γ* was calculated as *Γ* = *W*(*λ*_c_) *L*_0_, where *W*(*λ*) is the integral of the stress–stretch curve of the uncut sample and *λ*_c_ is the critical stretch when a fast fracture is observed. In the test, *λ*_c_ was defined as the stretch where the stress–stretch curve of the pre-cut sample reaches the peak (the displacement corresponding to the peak force) [19].

*Cyclic tensile testing*. Cyclic tensile tests were carried out at room temperature on PBS samples with a pre-stretch ratio of *λ*_fix_ = 7. The rate of loading and unloading were 5 mm min^−1^. The dumbbell-shaped sample was stretched to the set strain rate, then the tensile force was released at the same speed until the sample returned to the initial position. Hence the stretching-unloading process is considered as a cycle, and a series of stress–strain cycle curves were obtained by gradually increasing the tensile strain rate (*ε*_n_) to *n* times, and increasing the *ε*_n_ step by step until it exceeded the maximum strain of the sample to failure. The dissipative energy (*U*_n_) of each cycle was calculated from the area between the *n* loading and unloading curves, and the total dissipative energy (*U*_n_) from first to *n* was determined as the superposition of the single dissipation energy of 1–n. The total dissipate energy (*U*_n_) and work of tension (*W*_n_) of *n* times sample stretching-unloading were estimated by the following Equation (1–3) [22]:(1)$$U_{n}=\int_{0}^{\varepsilon_{n}}\sigma_{n}d\varepsilon -\int_{0}^{\varepsilon_{n}}\sigma_{n}^{’}d\varepsilon$$
(2)$$U_{n}=\sum_{i=1}^{n}U_{i}$$
(3)$$W_{n}=U_{n}+\int_{0}^{\varepsilon_{n}}\sigma_{n}^{’}d\varepsilon$$
where *ε*_n_ is the *n*th tensile strain rate, *σ*_n_ is the *n*th tensile stress, and *σ*’_n_ is the *n*th unloading stress. The ratio of dissipated energy to destructive work (*U*/*W*) in the destruction process indicates the irreversible work generated by internal failure in the tensile process. The larger the *U*/*W* value, the more serious the internal structure damage of the material.

#### 2.2.2. Crystallization Process and Crystalline Structure Characterization

*Crystallization Process*. The non-isothermal crystallization behaviors of the PBS samples were examined by differential scanning calorimetry (DSC) using a (DSC-8000, Perkin Elmer MA, USA) under a nitrogen atmosphere. For as-prepared chain-extended PBS and pre-stretched PBS (*λ*_fix_), maximizing the formation of further crystalline domains during the tensile induced crystallization. After testing, the sample was sealed in an aluminum crucible (about 4 mg) and tested in the temperature range between 30 °C and 180 °C at a heating and cooling rate of 10 °C min^−1^ under a nitrogen atmosphere with a flow rate of 30 mL min^−1^. The peak crystallization temperature was recorded during cooling and identified as the “standard” crystallization temperature (*T*_c_). In order to ensure the crystal structure integrity of the pre-stretched samples, it is not necessary to erase the thermal history except for with virgin PBS. Crystallinity is a mathematical statistics concept expressed as percentage of crystal (*χ*_c_). It can be calculated from the PBS melt content according to Equation (4), where Δ*H*_m_ is the melting enthalpy and $\Delta H_{m}^{0}$ is the melting enthalpy of fusion of the complete crystallization equal to 110.5 J g^−1^ [23].
(4)$$\chi_{c}\left[\%\right]=\frac{\Delta H_{m}}{\Delta H_{m}^{0}}\times 100\%$$

*X-ray Diffraction Scattering (WAXS and SAXS).* Wide-angle X-ray diffraction (WAXS) analysis was measured in a diffractometer at room temperature on a diffractometer (D/max 2500 VB2C/Pc, Panaco Instruments Co., Ltd, Netherlands) with CuKa X-ray radiation and a computerized data collection. The operating conditions of the X-ray source were set to 40 kV and 200 mA in the 2*θ* scan range of 5° to 90°. The data were normalized with respect to the incident beam intensity in order to correct for primary beam intensity fluctuations. The sample was irradiated with X-rays with a wavelength of 1.542 Å (*λ* = 0.1542 nm) as the radiation source. Notably, the X-ray profiles were recorded in the meridional direction. The prepared products were made into samples 1 mm thick. Small-angle X-ray scattering (SAXS) patterns were acquired at room temperature, operated with a 0.02 step size of 2*θ* from 0.5° to 10.0°. The absolute intensity for *I*(*q*) was determined using a four slit collimation system and the measurement of absolute intensity was carried out on standard samples. The scattering vector *q* was defined as *q* = 4π *λ*^−1^ sin *θ* with 2*θ* being the scattering angle. Raw SAXS and WAXS pattern data were processed with corrections by mathematic-based JADE software before analysis. 

#### 2.2.3. Morphological Characterization

*Scanning Electron Microscopy (SEM).* An SEM (SU8010, Hitachi Limited Co., Ltd, Tokyo, Japan) was used to characterize virgin and pre-stretched PBS sample morphology. Samples were coated with gold before analysis. Electron micrographs were taken with an acceleration voltage of 7.0 kV. In this study, we used concentrated sulfuric acid (30 mL), concentrated phosphoric acid (20 mL) and potassium permanganate (0.2 g) to etch the amorphous regions of PBS to observe its crystal morphology.

*Polarized Light Optical Microscopy (PLOM).* Spherulitic morphologies were observed by PLOM using a (Leica DM2500P, Weztlar, Germany) polarized microscope equipped with polarizers and a sensitive red tint plate (this was employed to determine the sign of the spherocrystal). A British Linkam hot stage connected to a liquid nitrogen system was used to control the temperature. The samples were pressed on a glass slide and covered with a glass coverslip. They were heated to a temperature of 130 °C (above the DSC melting peak). Similar to DSC testing, all samples except pre-stretched samples were kept at this temperature for 10 min to erase previous thermal history. Samples were then quickly cooled to the selected crystallization temperature 80 °C for 10 min to allow the crystals size to grow fully. Micrographs were taken with a Leica DC420 digital camera.

#### 2.2.4. Rheological Analysis

*Rotational rheometer*. The rheological behavior of PBS samples was investigated at 140 °C in dynamic mode using a rotational rheometer (DHR-2, TA-2 Instruments company, USA) with a parallel-plate (25 mm in diameter with a gap of 1.0 mm). The complex viscosity (*η*^*^), storage modulus (*G*′), and loss modulus (*G*″) were monitored at various frequencies. The frequency range was 0.1~100 rad s^−1^, and the maximum strain was fixed at 0.5% to ensure that these analyses were within the linear viscoelastic region under nitrogen. The real shear rates and zero-shear-viscosity were calculated using the Carreau-Yasuda model.

#### 2.2.5. Infrared Spectroscopic Analysis

*FT-IR Analysis*. FT-IR spectra were recorded at 25 °C and then subjected to thin film analysis using a Fourier-transform infrared spectrometer (Vertex 70, Bruker, Germany). The spectra were recorded in absorbance mode with a spectral resolution of 2 cm^−1^. PBS thin films were laid on a zinc selenium disk. Each spectrum was obtained within the range of 4000 ~ 500 cm^−1^.

## 3. Results and Discussion

### 3.1. The Influence of Chain Extension Reaction on PBS

PBS_A0.6_ and PBS_A0.6B0.8_ polymers were produced via chain extension of PBS–COOH with PBS–OH using ADR9 and BOZ as two-steps chain extenders at 140 °C for 15 min. The possible mechanism diagram is illustrated in Figure 1. In our previous work, we modified PBT with chain extenders ADR9 and BOZ [20]. The epoxy multifunctional groups react to have higher activity with carboxyl groups. The epoxy functional groups of ADR9 can be applied to the linear polymer chain of PBS to generate a product that includes a large number of short-branched chains. This product will form part of the hydroxyl groups which will react with BOZ, forming a kind of polymer with a long-branched, star-like molecular chain structure, which has high mechanical properties as well as processability.

We performed a series of measurements to demonstrate the effectiveness of the chain extension reaction in PBS. Gel fraction (GF) characterizes the degree of cross-linking and carboxylic terminal concentration group (CTCG) characterizes the degree of PBS–COOH or PBS–OH consumption during the chain extension reaction. These measurements are the most direct evidence for the chain extension reaction of PBS. It can be seen from Figure 2a, the GF of PBS_A0.6_ and PBS_A0.6B0.8_ samples increased with single component chain extension and collaborative chain extension. The GF of PBS_A0.6B0.8_ was 1.61%, which was not shown to have significantly affected the machining performance of PBS during the chemical chain extension reaction progress. The CTCG decreased to 28.2 mol t^−1^ for PBS_A0.6B0.8_ samples compared with 37.4 mol t^−1^ for virgin PBS samples. These results show that the end group (–OH and –COOH) concentrations of PBS were consumed by chain extension. Low-gel (low-crosslinked) content resulting from chain extension reactions often results in high mechanical properties. As shown in Figure 2b, the tensile strength of PBS_A0.6B0.8_ increased from 32.5 MPa (virgin PBS) to 45.1 MPa, while the elongation at break increased to 484%.

Molecular weight is closely related to viscosity; therefore, we measured the characteristic viscosity and average molecular weight of the polymer via Ubbelohde viscometer. Intrinsic viscosity ([*η*]) characterizes the branching and coupling of the molecular chain and the average molecular weight ([*M*_η_]) describes the change in the molecular weight after chain extension. Figure 2c shows that the intrinsic viscosity of PBS_A0.6B0.8_ increased from 0.71 dL g^−1^ to 1.19 dL g^−1^ compared with unmodified PBS, while the average molecular weight increased to approx. 40,000 g mol^−1^. These results indicate both ADR9 and BOZ were effective chain extenders for PBS. The relationship between complex viscosity (*η**) and angular frequency is shown in Figure 2d. Appendix A show shear thinning behavior occurred in the high frequency range. Compared with virgin PBS, the viscosity of the system increased after chain extension. 

Identical conclusions were obtained in infrared spectroscopic analysis shown in Appendix A. The observed variations confirm the occurrence of chain extension reaction between PBS and the chain extenders. We used DSC to measure the crystallization behavior of PBS before and after chain extension, shown in Figure 2e,f. The relevant data are presented in Appendix A. The crystallization temperature decreased with chain extension progress while the melting temperature increased. Moreover, after chain elongation, the regularity of chain segments increases, resulting in entanglement between chain segments at the molecular level for the gradual increase in crystallinity. To further elucidate the rheology character of PBS samples in a large angular frequency range [24], we used time temperature equivalence shift (TTS) data to obtain a wider *η*^*^ range and *G*′ and *G*″ curves in Figure 2g,h and Appendix A. Equation and method details can be found in the Appendix A. The Carreau-Yasuda model was used to fit the virgin PBS viscosity curve resulting in *λ* = 3.49 s and *η*_0_ = 515 Pa s. The modified Carreau-Yasuda model represents the best fit for the viscosity function of long-chain branched PBS-ADR9/BOZ samples. Two characteristic relaxation times were determined: *λ*_1_ = 9.21 s, *λ*_2_ = 0.5 s, and *η*_0_ = 1482.78 Pa s. These results show the long-chain branched structure formed after the two-step chain extension relaxation time was reduced. According to the analysis above, the addition of chain extender effects the molecular chain structure. 

To achieve a better understanding of how the chain extension reaction affect the crystal structure of PBS, Figure 2i illustrates the WAXS diffraction pattern of the samples. The samples show peaks at 2θ = 19.7° and 22.6°, corresponding to the diffraction peaks of the (320) and (130) crystal planes, respectively. The results of DSC and WAXS show that the crystal structure of PBS was not changed by chain extension.

### 3.2. The Energy Release Rate of PBS Samples

The energy dissipation mechanism of PBS can be further analyzed by studying the relationship between dissipated energy *U* and strain *ε*_n_, and the relationship between *U*/*W*_t_ and strain *ε*_n_. We used dumbbell-spline samples and performed cyclic tensile tests on both notched (Figure 3a,b) and unnotched samples (Appendix A). In this research, the excellent mechanical performance of these PBS samples were closely related to the effective energy dissipation, which can be demonstrated by hysteresis of the loading−unloading curves of the samples stretched to different maximum strain *ε*_1_, *ε*_2_ … and *ε*_5_.

As shown in Figure 3, large hysteresis was observed during the loading−unloading process of the chain extension sample, indicating the distinctive energy dissipation through the destruction of the interior structure under loading. The ratio of dissipated energy (energy release rate *U*_1_, *U*_2_ … and *U*_5_) and tensile failure work in the failure process (*U*/*W*_t_) was calculated as Equation (1 ~ 3). As *ε*_n_ increased from 5% to 30% of the chain extension sample, *U* increased from 0.15 to 1.9 × 10^6^ J m^−2^, while the dissipated energy *U* of pristine samples increased from 0.1 to 1.0 × 10^6^ J m^-2^ as *ε*_n_ increased from 2% to 22%, indicating that compared with the virgin sample, the gradual fracture of the internal structure of the reinforced sample requires more dissipate energy due to its high degree of molecular chain entanglement and branching. The curve of dissipated energy to tensile work *U*/*W*_t_ shows a near linear relationship, indicating the continuous structural destruction of the PBS during the cyclic loading−unloading process.

### 3.3. Characterization of the Crystalline Morphology in PBS Samples

We use the chain-extended sample (*λ*_fix_ = 1) and then pre-stretched them in a drying oven (accessories are provided by universal tensile testing machine) at 95 °C. Higher toughness of chain-extension modified PBS allows the modified sample to have a higher pre-stretching ratio than virgin PBS. As shown in Figure 4a and b (quantitative data are shown in Appendix A) both the pre-stretching and virgin PBS show distinctly endothermic peaks, with measured crystallinities of 35.6 (virgin PBS), 41.2 (*λ*_fix_ = 3) and 43.3 wt% (*λ*_fix_ = 5), respectively. Moreover, when the higher pre-stretching PBS is *λ*_fix_ = 7, the crystallinity of PBS increased to 45.7 wt% (Figure 4c). 

The increased crystallinity implies more crystalline domains nucleate during the chain extension reaction and pre-stretching process [25]. The crystallinity gradually increased further by increasing the pre-stretch ratio. When the sample was pre-stretched for *λ*_fix_ = 9 at 95 °C, the crystallinity reached 47.6 wt%. During the pre-stretching process, on account of the change of crystalline morphology of the polymer, the amorphous part undergoes stress-induced crystallization along the stress direction. To further elucidate the change of crystalline morphology, we measured the average distance between adjacent crystalline domains *d* using SAXS and average crystallite sizes perpendicularly across the planes *D* using WAXS. SAXS measurements on samples of pre-stretching PBS measured the scattering intensity [*I* (*q*)]^2^ versus the scattering vector *q* shown in Figure 4d, there is no other peak in the plot of intensity [*I* (*q*)]^2^ versus the scattering vector *q* for the PBS samples (Appendix A), which implies that neither the chain extension reaction nor the pre-stretching of PBS crystal generate a new crystal structure on the basis of the original PBS crystal morphology. The average distance between adjacent crystalline domains *d* can be calculated from the critical vector corresponding to the peak intensity *q*_max_, following the Bragg expression in Equation (5) [26].
(5)$$d=2\pi /q_{max}$$

To achieve a better understanding of the crystal morphology of PBS, we perform WAXS measurements on various PBS samples shown in Figure 4e, the corresponding diffraction peaks have strikingly different positions, distributions, and intensities of the (320) and (130) crystal planes and amorphous halo, respectively. As can be seen from Figure 2i and Figure 4e, the crystal structure of PBS is monoclinic crystal system, and its lattice parameters (*β* crystalline form) are as follows: *a* = 0.523 nm, *b* = 0.908 nm, *c* = 1.079 nm, and *β* = 123.78° [26].

In addition, small peaks at 2θ = 28.5° and 33.1° are also observed in the PBS samples. It is seen in Appendix A that the positions of the diffraction peaks of the samples did not change as the pre-stretching ratio was increased, indicating the crystal form of the samples had not changed. By identifying the half width of the maximum diffraction peak *β*, the average crystallite sizes perpendicularly across the planes *D* can be approximated using Scherrer’s Equation (6) [27],
(6)$$D=\frac{k\lambda }{\beta cos\theta }$$
where *k* is a constant varying with the actual shape (is related to FMHM) of the crystalline domain, *λ* is the wavelength of x-ray diffraction (1.542 Å), and *θ* a half of testing angle (2*θ*) from (hkl) plane. Here, *β* is the FMHM of the measured sample (double-line correction and instrument factor correction must be performed in radian), and the dimensionless shape factor *k* is set as 1, approximating the spherical shape of the crystalline domains. The average crystallite size is spread perpendicularly across the planes (*D*_hkl_). 

As shown in Figure 4f, by increasing the pre-stretch ratio from 1 to 9, the *D*_hkl_ decreased from 17.1 to 11.2 nm. This trend is consistent with the increase of *D*_hkl_ with different pre-stretch ratios, because orientation promotes the refinement of the crystal domain and induces crystallization for the amorphous polymer chain. Compared with the virgin PBS, the pre-stretched samples had a smaller average distance between adjacent crystalline domains *d* and average crystallite size perpendicularly across the planes *D*, and contains more crystals per unit volume (Figure 4d inset). As shown in Figure 4f, the average distance between adjacent crystalline domains for the *λ*_fix_ = 1 was calculated to be 45.2 nm in the chain extension state. As the pre-stretch ratio increased to 9 (*λ*_fix_ = 9), *d* increased to 53.5 nm. As a control case, we also measured SAXS profiles of the pristine sample (virgin PBS). The value of *d* in virgin PBS was approx. 30 nm, smaller than the distance in the pre-stretched sample (48 nm). These results indicate that in the process of stretching, the amorphous polymer chains and the crystal region of the polymer extend in the orientation of the stretching direction, resulting in crystal refinement and increasing the average distance between adjacent crystalline domains. The PBS spherulites deform and orientation along the stress direction, resulting in reduction of average crystal size. With the crystalline morphology changes along the stretching direction the regularity of chain segments increases by high orientation, there are more oriented crystals per unit volume, which improves the crystallinity of the sample, which is also confirmed in the crystallinity test results of DSC in Figure 4a,b.

PLOM was used to obtain phase images of the pristine sample and the pre-stretched samples. As shown in Figure 4g, large and clear spherulites morphology domains were observed in the pristine samples, while the pre-stretched sample shows thick and thin “stripes” in the crystalline domains. The phase images show that compared with the pristine sample, the crystal morphology of the pre-stretching sample was transformed from spherulite shape to elliptic crystal or extend-chain crystal.

### 3.4. Characterization of Flaw Sensitivity Properties of PBS

We used the single-notch method widely used in fracture tests to measure the flaw sensitivity of PBS. Notably, all fracture toughness tests in this study were performed on pre-cut samples with an initial crack 0.2 times its width. We tested the flaw sensitivity of both before and after chain-extended PBS samples by cutting an initial flaw ~0.8 mm on the edge of each sample, and uniaxially stretching the sample. 

In virgin PBS, the flaw quickly fractured throughout the entire sample once the sample stretching started (Appendix A ). In contrast, the crack propagation occurred along the loading direction in the chain extension sample with *λ*_fix_ = 1, leaving the sample fractured by crack propagation at a larger displacement. The low molecular weight part of PBS molecular chain decreased as a result of chain extension, which led to a decrease in notch sensitivity. The degree of anisotropy in PBS was tunable by the pre-stretch *λ*_fix_. For all chain extension samples with different values of *λ*_fix_, we observed a strengthened stress–stretch curve in the tensile direction. With chain extension in PBS and mechanical fixed *λ*_fix_, the tensile strength of the PBS sample increased by nearly 6 times reaching up to 185 MPa for 1 day fixed after pre-stretching in the stress–stretch curve (Figure 5a). By contrast, the stress–stretch curve of virgin PBS increased 4 times (Appendix A). Work of tension (*W_c_*) is an important parameter to characterize material toughness and is calculated from the integral area under the stress–stretch curve of uncut samples. Sample work of tension reached 1.8 × 10^8^ J m^−3^, which was higher in toughness than the 1.6×10^8^ J m^−3^ in virgin PBS (Appendix A). Characterizing the degree of anisotropy in terms of the fracture toughness when *λ*_fix_ > 2.0, the material is anisotropic and the crack will propagate along its initial direction under uniaxial loading.

When the fracture toughness of the material is approaching 10^6^ J m^−2^ (*λ*_fix_ = 5), the sensitivity of a soft, elastic material to flaw can be estimated by a critical length scale *Γ*/*W*_c_, where *W*_c_ is the work to rupture measured with no or negligible flaw. The elastic-toughness sample during its first-time loading had *Γ* ≈ 10^6^ J m^−2^ and *W*_c_ ≈ 10^8^ J m^−3^, leading to a critical flaw sensitivity length scale of *Γ*/*W*_c_ ≈ 0.01 m. In comparison to the without chain-extended PBS sample, *Γ* ≈ 2 × 10^5^ J m^-2^, *W*_c_ ≈ 10^8^ J m^−3^, and *Γ*/*W*_c_ ≈ 0.002 m (Appendix A). As the pre-stretch ratios increased, the flaw sensitivity critical length further increased. In particular, for the pre-stretch ratio 7 (*λ*_fix_ = 7), the flaw sensitivity critical length achieved 0.02 m. These results show the flaw sensitivity of PBS decreased after chain extension, indicating that pre-stretching improves the orientation of molecular chains, and the interaction between molecular chains is increased by increasing crystallinity. 

The chain-extension samples suffer a toughness fracture and gradual crack extension from an initial flaw under uniaxial tension. In addition to flaw sensitivity and stretch toughness tests, we also measured nominal stress versus stretch curves of all PBS samples to obtain their Young’s moduli and tensile strengths shown in Figure 6a,b, both the Young’s modulus and tensile strength increase with pre-stretching and show marked enhancements when the pre-stretching state reached *λ*_fix_ = 5. Compared with the pristine sample, the Young’s modulus and tensile strength of the chain-extension reinforced sample (*λ*_fix_ = 5) increased from 220 MPa up to 320 MPa and from 125 MPa up to 179 MPa, respectively. 

The notion of flaw sensitivity applies to all materials. We collected data of fracture toughness *Γ* and work of tension *W* of various materials, e.g., ceramics, polymers, biomaterials, metals, etc. [28] We plotted the data for various materials in a material space with *W* and *Γ* as coordinate axes (Figure 7). This serves to show that the length of flaw sensitivity has a large range, from nanometers for brittle materials to centimeters for tough materials. For non-inorganic materials like silica glass and alumina, measuring the fracture energy in the small-flaw limit is a difficult laboratory experiment and relative results are rarely reported because the small-flaw limit rupture of the cracks approach an infinitesimally tiny size (<10^−12^ m) [28]. By contrast, for elastomers or high toughness materials (e.g., polyethylene and natural rubber) and gels, the small-flaw limit is readily reached when the cracks are below millimeters. In practice, tough materials and gels can work in the small-flaw limit, the large-flaw limit, and anywhere in between. As we have commented before, the scatter of the rupture data measured using uncut samples is large for brittle hard solids, but small for tough materials and gels. In this work, the length of flaw sensitivity *Γ*/*W* of the strong and tough PBS is higher after chain extension and pre-stretching process.

### 3.5. Spherulite Morphology and Crystal Structure

Orientation has a significant effect on spherulitic morphology and spherulite size of crystalline aliphatic polyesters. As mentioned above, orientation has an effect on crystallization. Here, the effect of chain extension with pre-stretch on crystalline morphology was studied using PLOM. 

Both the spherulite morphologies of virgin PBS polyesters formed at the 75 °C crystallization temperature and the spherulitic morphologies of the pre-stretched state (with chain-extension) are shown in Figure 8, the virgin state and the self-nucleation process see Appendix A. The characteristic “Maltese Cross” extinction patterns of banded spherulites were observed despite the different states (virgin PBS and *λ*_fix_ = 1). The spherulites of virgin PBS show the spherulite diameter is about 50 μm before the spherulites impinge with each other. Nonetheless, the pre-stretching (*λ*_fix_ = 1) state developed negative spherulitic superstructural aggregates that resemble those of virgin PBS, but exhibit irregular edges (not perfectly circular). 

It is clear that the size of the spherulites shrunk, and the number of spherulites increase with chain-extension. These results show the difficulty of nucleation because of the molecular chain extension and increased degree of branching. As the pre-stretch ratio increased, the polyesters show deformation of spherulites (*λ*_fix_ = 3). To determine the change of the samples from folded-chain crystal (FCC) to extended-chain crystals (ECC) along the stretching direction were measured based on the condensed state. As shown in Figure 9, these crystalline morphology changes from FCC to ECC mechanism should influence the stretching scaling relationship. 

Through uniaxial stretching, we noted that the crystal morphology also deforms along the stress direction (by PLOM). It is further assumption that the crystalline morphology changes from FCC to ECC along the stretching direction. We carried out preliminary research on orientation of the crystalline regions using Raman spectrometry to analyze samples with different stretching ratios (*λ*_fix_ = 1 and 9, Appendix A). From the point of view parallel to the orientation direction (ZZ) and perpendicular to the orientation direction (XX), except that the peak intensity of 1720 cm^−1^ is the same, the residual peak strength ZZ is greatly increased compared with XX. It shows that the anisotropy of the crystal is obviously enhanced after the sample is taken. Comparing ZZ and XX, after applying vertically polarized light to the sample, the C-O-C peaks corresponding to 950, 1091 cm^−1^ and 1471, 1431 cm^−1^ of the sample have obvious changes and Raman shifts. Our research on orientation by 2D WAXS and related follow-up experiments are being carried out. 

Some spherulites are initially stretched into FCC (*λ*_fix_ = 5), and finally aligned into regular and complete ECC along the stretching direction. A more systematic study refers to the methods in the reference according to Michel levy diagram [35] on the mechanism of crystalline morphology transition is ongoing.

In addition, under the action of tensile stress, PBS samples are inclined, slipped and twisted to form crack chains under high orientation (*λ*_fix_ = 9), and the original spherulite structure is destroyed forming a “microfilament crystal” structure. Results of PLOM and SAXS are consistent with the significant influence of orientation on PBS spherulites. Tensile orientation has a significant effect on spherulitic deformation of PBS, which is consistent with the results of PLOM and SAXS. In order to further observe the morphology of PBS, the effect of orientation on the crystal structure of PBS was observed by SEM. As shown in Figure 9, the spherulite structure is obvious in virgin PBS (a—i) and chain-extended PBS (b—i). The spherulite size decreased after chain extension with the former measuring approx. 50 μm, while the latter is approx. 40 μm. Similar crystal orientation structure was also observed in (a)—(i–viii) and (b)—(i–viii) images. With the increased degree of orientation, a fibrous crystal morphology was observed (*λ*_fix_ = 9).

## 4. Conclusions

In summary, we have described a principle of flaw-insensitive materials under chain-extension and load through crack propagation. This work analyzed the flaw sensitivity of stretchable and high toughness materials. We measured work of tension *W* using uncut samples and fracture toughness *Γ* using samples containing large cuts (0.2 times its width). We identified a length of flaw sensitivity *Γ*/*W*. We have also proposed that the design principle for fracture toughness materials is to make the crack fracture of chain-extended PBS requiring energies per unit area much higher than that for fracturing a polymer chain of virgin PBS. We have demonstrated that the fracture toughness can be greatly enhanced by designing pre-stretch orientation in chain-extended PBS samples. We further confirmed the average size of crystalline domains decreased while the average distance increased with the tensile orientation process. The reported mechanism and strategy for designing anti-fracture and low flaw sensitive PBS can enhance the fracture toughness performance of PBS making a number of future research directions of the extrusion product applications possible.

## Figures and Tables

**Figure 1 polymers-14-01876-f001:**
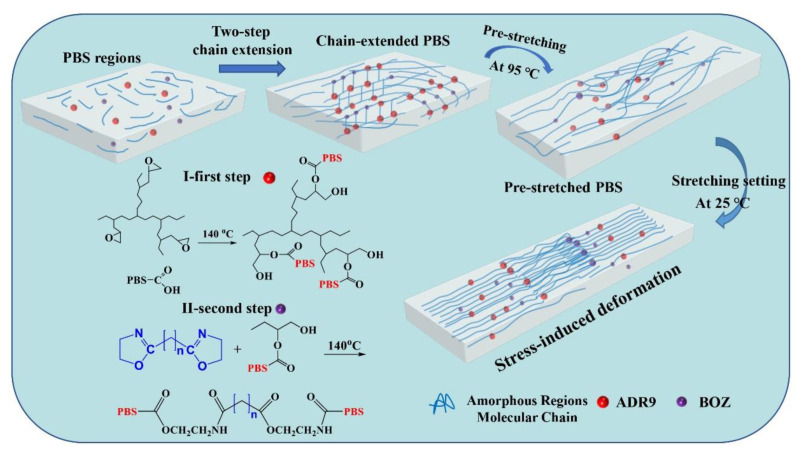
Mechanism diagram of collaborative chain extension of PBS.

**Figure 2 polymers-14-01876-f002:**
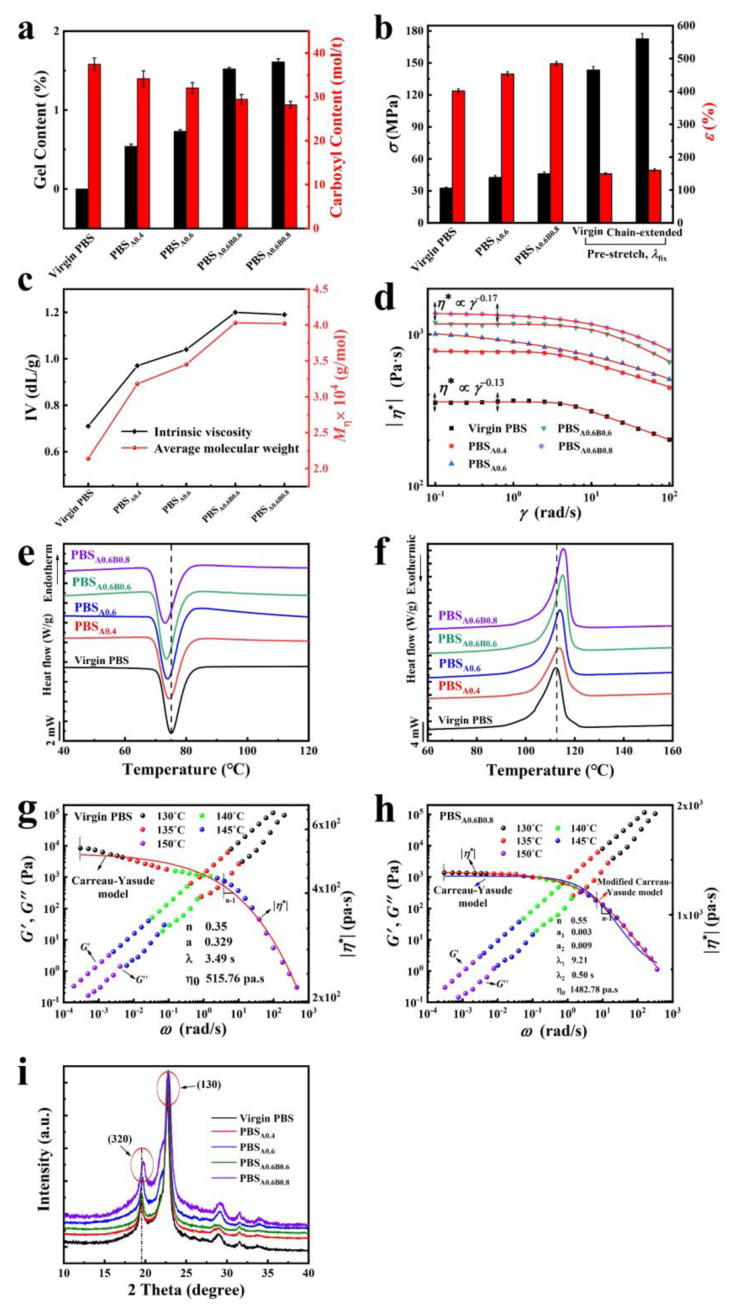
Measurement of two-steps chain extension factors of PBS. (**a**) Variation of gel fraction and carboxyl group content of PBS samples. (**b**) The tensile strength and elongation at break of chain-extended PBS were obtained at a rate of extension of 20 mm min^−1^ at room temperature. (**c**) Intrinsic viscosity and average-molecular weight of PBS samples. Non-isothermal crystallization curves for PBS samples at 140 °C. (**d**) Double logarithm diagram of complex viscosity changing with angular frequency at 150 °C and 1 ~ 100 rad s^−1^. (**e**) Cooling curve (10 °C min^−1^), (**f**) heating curve (10 °C min^−1^). Viscosity function (*η**), storage (*G*′) and loss modulus (*G*″) of (g) virgin PBS and (**h**) long-chain branched PBS_A0.6B0.8_ with best fits based on the Carreau-Yasuda model. Detailed measure methods can be found in the Appendix A. (**i**) WAXS patterns of PBS samples by fitting in Jade with 2θ from 5° to 90°.

**Figure 3 polymers-14-01876-f003:**
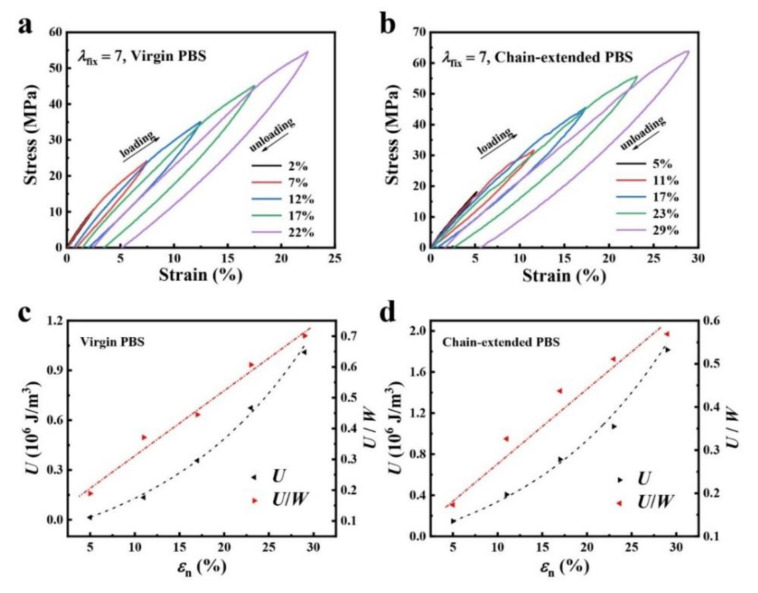
Cyclic tensile and energy release rate of PBS samples. Hysteresis curves of virgin samples (**a**) and chain extension states (**b**) with various strains and cyclic tensile loading–unloading curves. Virgin PBS and chain-extended PBS samples were stretched five times in cycles before the maximum critical strain *ε*_n_, recorded as *ε*_1_, *ε*_2_ … and *ε*_5_. The corresponding dissipated energy (*U*) of one loop at different maximum strain *ε*_max_, recoded as *U*_1_, *U*_2_ … and *U*_5_. The relationship between the ratio of dissipated energy to tensile work *U*/*W*_t_ and strain of the virgin state (**c**) and chain extension state (**d**). The loading and unloading stretch rates were 5 mm min^−1^.

**Figure 4 polymers-14-01876-f004:**
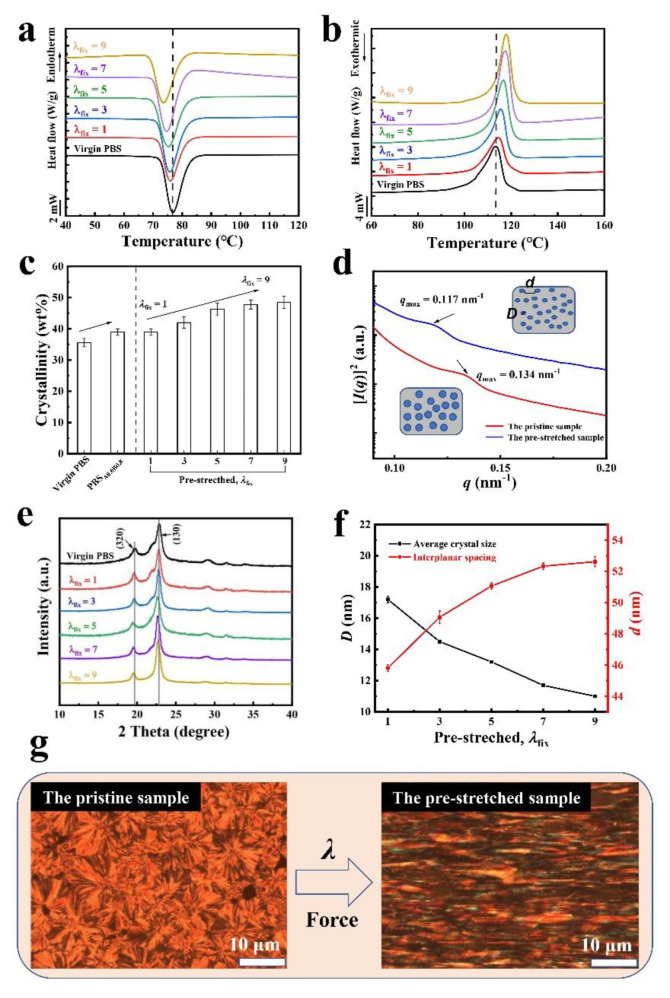
Characterization of crystalline domains in PBS. (**a**,**b**) Representative DSC curves for crystallization and melting of the various pre-stretching states, with pre-stretch ratios. (**c**) Measured crystallinity in virgin PBS, chain-extended PBS and pre-stretching states with pre-stretch ratios. (**d**) SAXS profiles of the pristine sample and the pre-stretching sample with q_max_ values extracted from the [I(q)]^2^ ~ q curve. (**e**) Representative WAXS profiles of virgin PBS and various pre-stretching states with pre-stretch ratios of λ_fix_ = 1, 3, 5, 7 and 9. (**f**) Estimated the average distance between adjacent crystalline domains *d* and average crystallite sizes perpendicularly across the planes *D* of pre-stretching states with pre-stretch ratios. (**g**) PLOM phase images of the pristine state and the pre-stretching state (scale bar = 10 μm).

**Figure 5 polymers-14-01876-f005:**
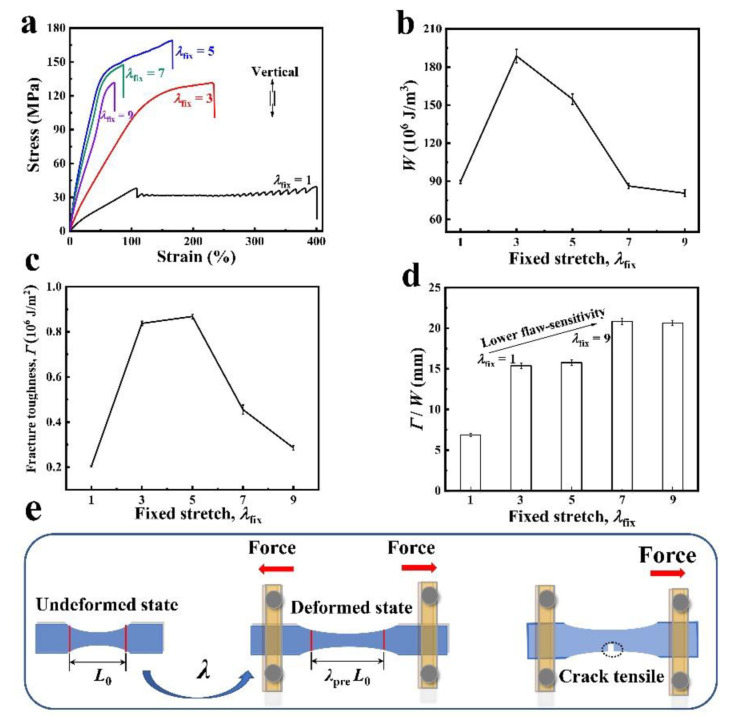
The analysis of the effect on fracture toughness prescribed by *λ*_fix_ of chain-extended PBS. (**a**) The stress–strain curve shows the sample becomes stiffer in the aligned direction with increasing *λ*_fix_. (**b**) The work of tension *W* of the sample is the integral of stress–strain curve with a pre-stretch ratio. (**c**) The relationship between the fracture toughness *Γ* and pre-stretch ratios. (**d**) The critical flaw sensitivity length scale *Γ*/*W* of the sample becomes a measure of lower notch sensitivity with increasing *λ*_fix_. (**e**) The PBS samples was pre-cut with an initial crack 0.2 times the width of the sample. An initial flaw (≈ 0.8 mm) in an unaligned pristine sample quickly propagates throughout the sample once the tensile spline is stretched, while the chain extension sample is slowly pulled apart. The notched tension of the virgin sample and the chain extension sample (*λ*_fix_ = 1) can be observed in Appendix A.

**Figure 6 polymers-14-01876-f006:**
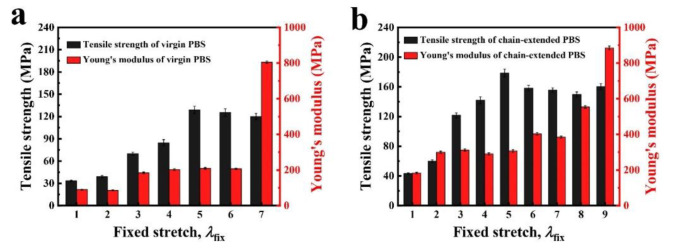
The Young’s moduli and tensile strengths of PBS samples. (**a**) Young’s moduli and tensile strength versus fixed stretch (*λ*_fix_) in virgin PBS. (**b**) Young’s moduli and tensile strength versus fixed stretch (*λ*_fix_) in chain-extended PBS.

**Figure 7 polymers-14-01876-f007:**
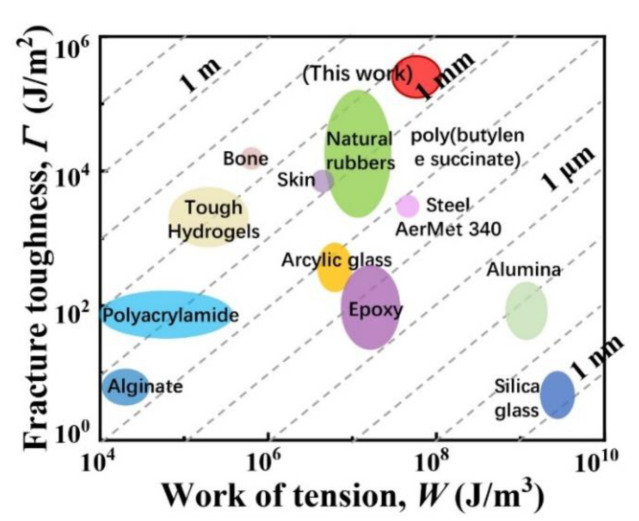
A space of material properties with fracture toughness *Γ* and the work of tension *W* on the axes. Also included are the slashes of constant values of the length of flaw sensitivity *Γ*/*W*. The stretchable materials in the current work are compared with other materials, e.g., natural rubbers [29,30], polyacrylamide hydrogels [30], alginate hydrogels [30,31], and tough hydrogels, [32,33], as well as steel, aluminum, bone, human skin, acrylic glass, epoxy, aluminum oxide, and silica glass [34].

**Figure 8 polymers-14-01876-f008:**
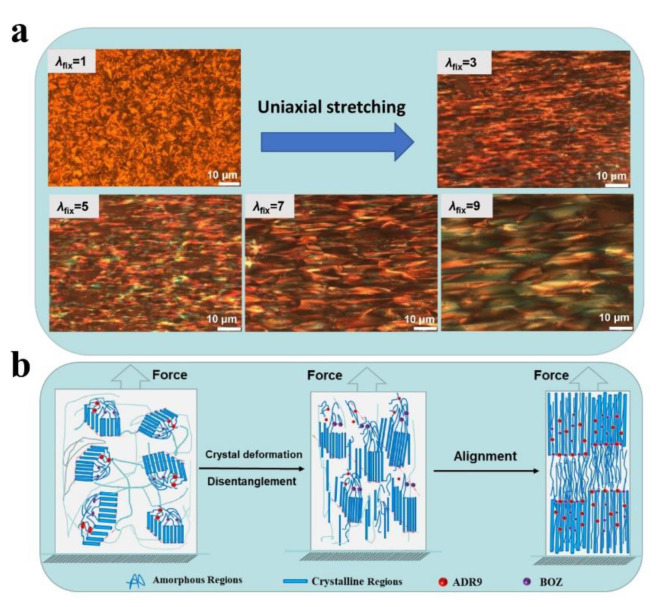
Polarizing microscope analysis of stretch oriented samples. Polarized light optical micrographs of the pre-stretched samples. (**a**) Micrograph of the sample surface taken in the pre-stretching state. The scale bar is 10 μm. All the samples were oriented with pre-stretch at 95 °C. (**b**) After uniaxial stretching, the crystal deformation and molecular chains are disentangled, and the molecular chains show anisotropy along the stretching direction.

**Figure 9 polymers-14-01876-f009:**
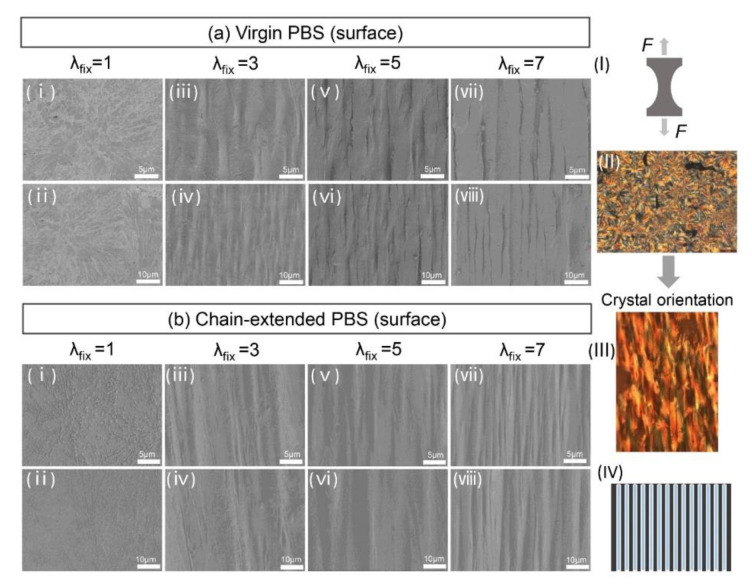
Scanning electron microscope analysis of orientated PBS before and after chain extension. The surface of virgin PBS (**a**)—(i–viii) and chain-extended PBS (**b**)—(i–viii) has as oriented spherular structure, (i) uniaxial tensile orientation diagram, (ii–iii) PLOM of spherulite structure change after crystal orientation, (iv) structural diagram of the fibrous crystal.

**Table 1 polymers-14-01876-t001:** Sample type, name abbreviation, and characterizations.

Sample Code	Chain Extension for PBS ^a)^			Pre-Stretching ^b)^			
	PBS	ADR9	BOZ	*λ* _pre_	*λ* _fix_	*ε* (%)	*L*_0_ (mm)
Virgin PBS	100	0	0	1.2	1	120	20
PBS_A0.4_	99.6	0.4	0	3.2	3	320	20
PBS _A0.6_	99.4	0.6	0	5.3	5	530	20
PBS _A0.6B0.6_	98.8	0.6	0.6	7.5	7	750	20
PBS _A0.6B0.8_	98.6	0.6	0.8	9.8	9	980	20

^a)^ Samples names correspond to the mass ratio (wt%) of ADR9 (A) and BOZ (B). ^b)^ All pre-stretching samples are based on the two-step chain-extended PBS_A0.6B0.8_. Notably, the toughness of PBS is enhanced due to chain extension, the sample can be stretched to 9 times the original gauge length *L*_0_, while the sample of virgin PBS (without chain extension) can be stretched to 7 times the original gauge length *L*_0_.2.3. Experimental Methods and Measurements.

## Data Availability

The raw/processed data required to reproduce these findings cannot be shared at this time as the data also forms part of an ongoing study.

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
