# Peer review of "High Mechanical Properties of Stretching Oriented Poly(butylene succinate) with Two-Step Chain Extension"

_polymers, 2022, doi:10.3390/polym14091876_

Round 1
Reviewer 1 Report
The reviewer carefully read the submitted manuscript entitled “High Mechanical Properties of Stretching Oriented Poly(butylene succinate) with Two-Step Chain Extension”. The authors have modified poly(butylene succinate) (PBS) by two-step chain extension using two kinds of extenders, ADR9 and BOZ and have studied the structure, morphology, fracture toughness and flaw sensitivity length scale of the chain extended PBS with various pre-stretch ratios.
Some of the experimental results show that the two-step chain extension method makes PBS have relatively high toughness. These results seem to meet the criteria of the journal “Polymers”. However, there are some problems in the submitted manuscript. The reviewer recommends the publication of the manuscript to the journal “Polymers” when all the problems listed below are resolved.
-Major Problem-
- Figure 4(f) shows that the average crystalline size D decreases and the inter-planar spacing d increases with the pre-stretched ratio λfix. These tendencies indicate that the crystallinity decreases with λfix. Figure 4(c), however, shows that the crystallinity increases with pre-stretched ratio λfix. What does this discrepancy come from?
- In Section 3.5, the authors have suggested that the crystalline morphology changes from folded-chain crystal (FCC) to extended-chain crystal (ECC) along the stretching direction by high orientation (λfix = 9). What is the evidence of the crystalline morphology for λfix = 9 is FCC? Figure 4b shows that the melting temperature increases with λfix. But this increase does not seem enough to show the change from ECC to FCC.
- The reviewer cannot understand Equation (3). The review think that σn+1 should be changed to σ'n. The term “the total dissipative energy (ΔUn)” at page 4, lines 160 and 162 should be changed to “the total dissipative energy (Un)”.
-Minor Problem-
- The characters in the figures (especially Figures 2, 4, and 5) are too small to read.
- This manuscript is very difficult to read. At least the text should be divided into paragraphs of appropriate length.
Reviewer 2 Report
The submitted manuscript presents a big and interesting work. The results are well described. However, I’d like to make some recommendations:
- Lines 226-227. The following sentence needs to be rephrased: “Moreover, due to the regularity of chain segments increases after chain extension for the gradual increase in crystallinity.”
- Line 271. Please specify in detail what the term used means: "interplanar distance d". Typically, the interplanar distance d is the distance between any planes in the crystal lattice. However, this distance can be measured using the WAXS method, not the SAXS method. This point should be properly described to avoid any misunderstanding.
- WAXS investigations: It is not clear in which direction the WAXS and SAXS profiles shown in Figures 2 and 4 were recorded with respect to the film stretch direction. They should be recorded in the meridional and equatorial directions, i.e. parallel and perpendicular to the film axis. Besides, it would be very useful to get the WAXS profiles of the azimuthal distribution of the x-ray intensity in order to obtain a quantity information about orientation degree of the crystalline regions.
- If the Authors calculate the crystalline sizes D, they should indicate the type of the crystal lattice of the given polymer and the unit cell parameters.
- Figure 6. Rebuild the graphs in the same ranges along the Y axis both in Figure (a) and Figure (b). This will make the difference in mechanical properties between virgin PBS and chain-extended PBS films more apparent.
Round 2
Reviewer 2 Report
- Lines 287-289. I would suggest to make one correction to make the sentence more clear: “Moreover, after chain elongation, the regularity of chain segments increases, resulting in entanglement between chain segments at the molecular level for the gradual increase crystallinity.”
- Please, add to the description of the WAXS and SAXS experiment the phrase that the X-ray profiles were recorded in the meridional direction.
- The Authors wrote in their reply: “About the orientation degree of the crystalline regions of PBS samples, both 2D WAXS (the azimuthal distribution) and polarized light microscope have been proved to be effective methods to obtain the degree of orientation.”
I believe this statement is not entirely true. Based on the azimuth data of the WAXS, it is possible to judge precisely the degree of orientation of the crystalline regions, while using the POM it is more difficult to make a clear distinction between crystalline and amorphous regions. Moreover, according to the WAXS data, it would be possible to obtain a quantitative assessment of the degree of orientation. In this article, the authors do not give any quantitative estimates of the degree of orientation, but only visual images of POM and SEM. - Authors wrote: “As can be seen from Figure 2i and Figure 4e, the crystal structure of PBS is monoclinic crystal system, and its lattice parameters (lattice constant) are as follows: a ≠ b ≠ c, α =γ= 90°, β ≠ 90°.”
These are NOT lattice parameters! This is just a general description of the monoclinic syngony. The lattice parameters are the exact values of a, b, c and β. Please specify them exactly!
